# Seasonal Variability of Cultivable Nitrate-Reducing and Denitrifying Bacteria and Functional Gene Copy Number in Fresh Water Lake

**DOI:** 10.3390/microorganisms12030511

**Published:** 2024-03-02

**Authors:** Jörg Böllmann, Marion Martienssen

**Affiliations:** Chair of Biotechnology for Water Treatment, Brandenburg Technical University Cottbus-Senftenberg, 03044 Cottbus, Germany; martiens@b-tu.de

**Keywords:** denitrification, nitrite reductase, MPN, vbnc state, PCR, microscopical cell number, dimictic lake, epilimnion, hypolimnion, sediment

## Abstract

This study describes the seasonal course of denitrifying and nitrate-reducing bacteria in a dimictic mesotrophic lake (Lake Scharmützelsee, Brandenburg, Germany) within a three-year period from 2011 to 2013. The bacterial cell numbers were quantified by the fluorescence microscopy, most probable number (MPN) and PCR-dependent quantification of the chromosomal 16S rDNA and of the *nirS* and *nirK* gene copy number. The highest seasonal differences (up to three orders of magnitudes) have been measured using MPN in the epilimnion. This variation was not reflected by PCR-dependent approaches or direct microscopical enumeration. At adverse conditions (low temperature and/or low nitrate concentrations), the differences between MPN and gene copy numbers increased by up to five orders of magnitudes and decreased to one magnitude at favourable environmental conditions. These results can be explained best by an increasing ratio of viable but not cultivable (VBNC) cells or dead cells at impairing conditions. In the hypolimnion, the courses of MPN and *nir* gene copy numbers were similar. This can be explained by a higher feeding pressure and therefore smaller amounts of dormant cells. In the pelagial in general, the total cell numbers enumerated by either microscopical or molecular approaches were similar. In the sediment, more than 99% of the DNA was obviously not related to viable bacteria but was rather DNA in dead cells or adsorbed to particle surfaces.

## 1. Introduction

Urbanization and human land uses such as agriculture have resulted in increased levels of bioavailable nitrogen in lakes, streams and rivers. Denitrification is the most important method of nitrogen elimination in limnic ecosystems. The determination of the abundance, function and activity of the responsible group of bacteria is an invaluable aspect of microbial ecology (ref. [1]) and can be a useful approach to estimate in situ turnover rates refs. [2,3,4] parallel to direct in situ measurements, which, however, are often difficult [5].

Many studies on denitrification rates and the responsible bacterial populations have been carried out mainly in marine and coastal ecosystems and especially estuaries (e.g., [6,7,8]), but samples were often taken only a few times per year [9] and were eventually analysed either using cultivation-based techniques [10] or molecular methods [9,11,12,13,14]. Although fresh-water ecosystems can serve as an important nitrogen sink in temperate regions, only little is known about the seasonal variability of the denitrifying bacterial populations in these systems [10]. The same is true for the relative impacts of the pelagial and sediment to the overall denitrification capacity. One reason could be the lack of adequate enumeration techniques for the different groups of bacteria in the nitrogen cycle and the many drawbacks of individual quantification methods.

Although several methods are available, the exact enumeration of denitrifying bacteria is still difficult [14]. Cultivation-based techniques have been applied for decades for the selective counting of functional bacterial groups. Their advantages and disadvantages have been often discussed [15]. They might only catch a small fraction of the bacterial group of interest [16,17], and are time and material consuming [18,19] and quite imprecise [20]. It is argued that cultivation-dependent enumeration will most likely underestimate the number of bacteria by magnitudes compared to molecular techniques [15,16,21,22]. With advances in molecular techniques, the selective detection and quantification of denitrifying bacteria based on the PCR of the genomic DNA of enzymes of this metabolic pathway became possible [11,13,23,24]. PCR-based methods are faster with less material effort required and are more precise, especially when using q-PCR or even more advanced molecular techniques. However, molecular techniques encounter several difficulties as well [25,26,27,28,29]. A major disadvantage in ecological means is that PCR techniques based on ribosomal or genomic DNA cannot distinguish between inactive or non viable cells or DNA in dead cells and metabolically active cells [24,30,31,32,33], but deliver a proportion of total cell counts, copy number or relative signal intensities rather than an absolute number of cells or biomass [1]. Only in systems with fresh biomass, the MPN counts and relative numbers of gene copies are closely correlated [34]. Thus, several authors concluded that molecular approaches are not sufficiently discussed in comparison with traditional methods [35]. Comparing cultivation-based and molecular cell-counting techniques and correlating their results with in situ activities is still difficult. Although [5,36] found a correlation between gene copy number and in-stream denitrification activity; ref. [14] suggest the use of gene abundance mainly as an indication of potential activity.

It has been shown previously that bacteria can enter a so-called viable but non-cultivable state (VBNC state) [37] as a response to harsh environmental conditions such as nutrient starvation, extreme temperatures, sharp changes in pH or salinity or oxygen availability [38]. In a lab study with a pure *Pseudomonas aeruginosa* strain, less than 0.1% of all bacteria were cultivable under nitrate starvation, whereas optimal growth conditions resulted in nearly identical cell counts using MPN and molecular techniques [24]. The discrepancies between MPN and molecular techniques may indicate the portion of bacteria that are not involved in the substrate-turnover processes. Especially in environmental systems with changing growth conditions, molecular-based techniques might not reflect the variability of active cell abundance [24].

Thus, the understanding of the nitrogen turnover and the study of the succession and the interrelationships among and between bacterial populations and their environment is still incomplete [39] and needs adequate enumeration techniques for metabolically active bacteria [29].

In this study, the abundances and seasonal variations of nitrate-reducing bacteria in the pelagial and upper sediment of a mesotrophic dimictic fresh-water lake (Lake Scharmützelsee, Germany) have been followed over a period of more than two years. Because of the difficulties in bacterial enumeration, we used and compared the cultivation-based MPN technique with the molecular quantification of the nitrite reductase encoding *nirS* and *nirK* gene copy number and 16S rDNA as well as with total concentrations of bacterial cells counted by filtration and fluorescence microscopy. The seasonal course is correlated and explained with the seasonality of the main growth parameters of denitrifying bacteria. The results show that the abundance of denitrifying bacteria is highly dynamic, which needs to be considered for the temporal and local characterization and quantification of denitrifying processes in those environments. Our results might improve the estimation of the denitrification capacities in comparable systems and their different compartments, could help to identify changing limitations and can improve the prediction of future developments [39] or lake-management activities.

## 2. Materials and Methods

### 2.1. Sampling Site and Physical and Chemical Parameters

The lake Scharmützelsee is a well-studied representative of a dimictic mesotrophic lake of the type LCB 1 of the European intercalibration lake type in the Central Baltic region [40]. It is located in NE Germany (52°15′ N, 14°03′ E), has a size of aprox. 12 km^2^, a mean depth of 8.8 m, a maximal depth of 29.5 m and a mean residence time of 16 years [41]. The depth of the thermocline varies between 6 and 10 m.

In the dimictic Lake Scharmützelsee, the environmental parameters temperature and oxygen and nitrate concentrations show typical annual courses with high similarities between the three studied years (Figure 1). Temperature was slowly increasing during summer and declined after mixing in fall. Oxygen dropped from high concentrations in winter to zero in the hypolimnion after stratification during spring. The nitrate concentrations showed distinct annual courses as well with peak concentrations in winter, a small decline in spring, another peak in early summer and depletion in fall.

In more detail, the water temperature was lowest in winter at down to 1 °C (Figure 1a). After stratification in spring, temperature in the epilimnion increased to up to 22 °C. In the hypolimnion, the temperature increased much slower and reached a maximum temperature of 10 °C in fall (Figure 1a).

The oxygen in the epilimnion and the mixed water column showed a seasonal course with maximum values in late winter and early spring (13. 7 mg L^−1^ in March 2011, 14.5 mg L^−1^ in February and March 2012) and minimum values in fall (5.8 mg L^−1^ in 2011 and 6.3 in 2012, Figure 1b). In the hypolimnion, oxygen dropped very fast to zero in late June after stratification and remained at zero until circulation in fall (Figure 1b).

The nitrate concentration showed a strong seasonal course as well (Figure 1c) with maximum concentrations (about 400 µg∙L^−1^) always in winter. Concentrations are displayed in logarithmic scale to better visualize low concentrations. In the epilimnion, nitrate dropped down in spring 2011 to 11 µg L^−1^ and reached a minimum value in July of 1.9 µg L^−1^ (Figure 1c). In 2012, it dropped to 1.2 µg L^−1^ in May and remained below 14 µg L^−1^ during summer. In 2013, nitrate concentration decreased to 2.3 µg L^−1^ in June and fluctuated between lower and higher values up to 40 µg L^−1^ in summer (Figure 1c). In fall and especially after mixing, nitrate concentration increased rapidly to the winter maximum (Figure 1c). In the hypolimnion, nitrate decreased in spring as well but remained at higher values compared to the epilimnion (64, 48 and 92 µg∙L^−1^ in 2011, 2013 and 2013, respectively). In summer, an additional short maximum in June or July of up to 300 µg∙L^−1^ could be observed (Figure 1c). Until October, it dropped down to minimum concentrations of around 3 µg∙L^−1^ in 2011 and 2012 and 1.7 µg∙L^−1^ in 2013, before it rapidly increased after mixing to maximum concentrations in winter (Figure 1c). Dissolved organic carbon (DOC) as another major parameter for denitrifying bacteria varied between 6 and 8 mg∙L^−1^ in the epilimnion with two exceptions in May 2012 (14 mg∙L^−1^) and October 2012 (4.4 mg∙L^−1^). In the hypolimnion, DOC ranged in 2011 between 7 and 10 mg∙L^−1^ without a noticeable seasonality. In 2012, the values decreased from June to October from 9.8 to 3.7 mg∙L^−1^.

### 2.2. Sampling

Pelagic samples were taken in the years 2011, 2012 and 2013 at the long-time monitoring site Rietz (RIE, 29 m depth) as pooled water samples of the entire mixed water column or separately from epi- and hypolimnion during stratification. Limnological parameters were measured and kindly provided by the Chair of Freshwater Conservation (Brandenburg University of Technology). Dissolved oxygen and temperature profiles were measured with a multiparameter probe (HYDROLAB H20, Hydrolab Corporation, Austin, TX, USA) connected to a field computer (HUSKY Hunter, Itronix Corp., Spokane Valley, WA, USA) in 0.5 m intervals. Mean values for the seasonal courses were calculated either for the epi- and hypolimnion separately (during stratification) or for the entire mixed water column (thereafter discussed together with epilimnion data). Nitrate and nitrite were analysed as a combined parameter according to standard methods (DEV 1976) as described by [41]. Nitrate and nitrite concentrations were summarized to NO_3_-N, since NO_2_ concentrations were always negligible. Dissolved organic carbon (DOC) concentrations were either provided by the Leibniz Institute of Freshwater Ecology and Inland Fisheries or were measured using a DIMATOC^®^ 100 (DIMATEC Analysentechnik GmbH, Essen, Germany) according to [42].

Intact sediment cores were taken at the same site according to [43] with an Uwitec^®^-corer (flutter valve, acrylic glass tubes, i.d. = 57 mm) parallel to the water samples and sliced into the first and second cm layer. The dry weight of the sediment at this site was between 5 and 6.5%. The portion of organic materials (oTS, determined as loss of ignition at 550 °C) was between 24% and 30% of the dry weight. Water and sediment samples were cooled at 4–6 °C and analysed within 12 h.

### 2.3. Total Cell Counts

The total amount of viable bacterial cells was counted by fluorescence microscopy (Nikon Eclipse LV 100, Nikon Vision Co., Ltd., Tokyo, Japan, with NIS Elements BR, Nikon Systems Inc, Tokyo, Japan) at 20- and 40-fold magnification. In total, 1 mL of a water sample or 0.5 to 1 mL of an appropriate diluted sediment sample (usually from the 10^−3^ dilution step of the MPN dilution series) was incubated with 100 µL of a Syto 9- Propidium iodide mixture (LIVE/DEAD^®^ BacLight™, 0.33 mM and 2 mM, Invitrogen AG, Carlsbad CA. USA) and filtered through a counting filter (Isopore Membrane Filters, 0.2 µm GTBP, Merck Millipore Ltd. Carrigtwohill, Irland) using a filtration unit (Sartorius, Göttingen, Deutschland) according to [24,44]. The filter was directly used for cell counting.

### 2.4. Most Probable Number

The number of denitrifying and nitrate-reducing bacteria in water and sediment samples was measured as the most probable number (MPN) according to [45] from one tenfold dilution series per sample with three tubes per dilution step in Nutrient Broth II (Sifin, Berlin, Germany) with 1.5 mg L^−1^ KNO_3_ in Hungate tubes. Durham tubes were added for gas capture. Gas production and decrease or loss of nitrate, analysed according to [46], confirmed denitrification. Nitrate reduction was confirmed by Nitrite accumulation, measured using the Griess-Ilosvay method (ref. [47]) without gas production. MPN and 95% confidence intervals were calculated according to FDA recommendations [48].

### 2.5. Molecular Approach for Microbial Enumeration

For the estimation of the total bacterial cell number and the number of denitrifying bacteria, a semi quantitative PCR method described by [24] was used, which correlates the amount of PCR products with a calibration curve generated with known cell numbers.

For DNA extraction, 50 mL of pelagic samples were filtered as mentioned above to obtain enough PCR product for quantification. Filters and native sediment samples were stored at −20 °C until extraction. Cells from filters were resuspended with purified water. A total of 1.5 mL of a 1:100 dilution of the defrosted sediment was centrifuged at 12,000× *g*, and the pellet was used for DNA extraction. DNA was extracted through several application steps of extraction buffer/chloroform and isopropanol according to [24]. The DNA samples were resuspended with 100 µL of purified water and stored at −80 °C. As a control for a constant extraction and PCR efficiency, DNA of a known cell number of *Pseudomonas aeruginosa* ATCC 27853 was extracted and amplified in parallel.

The chromosomal 16S rDNA gene was amplified using the universal primer set 27f (5′-AgAgTTTgATC(A/C)TggCTCA-3′) and 1525r (5′-AggAggTgATCCAgCC-3′). For enumeration of denitrifying bacteria, the chromosomal DNA of the nitrite reductase *nirS* and *nirK* genes were amplified with the primer set cd3aF (5′-gT(C/G)AACgT(C/G)AAggA(A/G)AC(C/G)gg-3′) and R3cd (5′-gA(C/G)TTCgg(A/G)Tg(C/G)gTCTTgA-3′ [49]. A PCR protocol described by the authors of [24] was used. Agarose gels were stained with enthidium bromide and the amount of PCR products was quantified with the software GelixOne G230 2.7. (biostep^®^, Jahnsdorf, Germany) in comparison with a quantification marker (MassRuler DNA Ladder Mix, Thermo Scientific, Schwerte, Germany). The initial number of gene copies was calculated with a calibration curve, which was established with known cell numbers (microscopical cell counting after cell staining with Syto 9) of *P. aeruginosa* ATCC 27853, which represents a widespread group of denitrifying bacteria. *nirS* and *nirK* gene copy numbers were summarised to *nir* gene copy number.

### 2.6. The Statistical Variance of the Different Methods

The MPN method is known to have a high inherent statistical error. According to [50], MPN values were considered significantly different when the 95% confidence intervals did not overlap, which is in most cases a difference of roughly two log units (least significant difference =1.01). In a previous study [51], the variance of the different methods was tested with five parallel samples of a *P. aeruginosa* culture at 4 different cell densities each. The standard deviation of the log-transformed microscopic counting was 0.05 and for the MPN result it was 0.41. The enumeration by PCR had a standard deviation of 0.15 log units generated from the same samples, which were extracted and amplified separately. Using the Student’s t statistics, a difference of 1.8 log units for MPN values, 0.22 log units for microscopic counting and 0.6 log units between two gene copy numbers can be considered significant (*p* = 0.05).

For correlations between the log-transformed cell numbers of the different enumeration approaches and between cell numbers and environmental parameters, the correlation coefficient (R) of Spearman was calculated using the software Origin Pro 8.5 (Origin Lab Corporation, Northampton, MA, USA).

## 3. Results

Total cell counts, numbers of cultivable nitrate-reducing and denitrifying bacteria, as well as the abundances of the chromosomal 16S rDNA gene copies and the sum of *nirS* and *nirK* gene copies, have been documented over the annual courses of three consecutive years from 2011 to 2013.

### 3.1. Epilimnion

In the epilimnion, the total microscopic cell counts ranged from 5 × 10^5^ to 7 × 10^6^ mL^−1^ (Figure 2a). Besides a small decline in summer 2012 and an increasing tendency by one magnitude from fall 2013 to spring 2014, there was no obvious seasonality.

The total cell numbers in the epilimnion calculated from the 16S rDNA gene copy number varied between 1.5 × 10^5^ and 2 × 10^7^ mL^−1^ (Figure 2a) and were comparable to the microscopic cell counts. In 2012, a slight increase from spring to summer by one magnitude was calculated, followed by lower values in fall and winter. The data suggest another decline in winter 2013–2014 from 2 × 10^7^ to 2.6 × 10^6^ mL^−1^. However, as shown for the microscopic cell counts before, the combined results from 2011 to 2014 did not reveal an obvious seasonality in the annual courses for the 16S gene copy numbers as well (Figure 3a). Both methods detected in average the same amount of bacterial cells (difference = 0.1 log units), although the 16S rDNA cell number showed a bit more fluctuation most likely caused by the higher methodical error.

Compared to the total microscopic and PCR-based cell counts, a strong seasonal variation was observed in the MPN of nitrate-reducing and denitrifying bacteria (Figure 2b). From April to August 2011, the MPN of nitrate-reducing bacteria raised from 1.1 × 10^2^ to 3.8∙10^3^ mL^−1^ and dropped down to 9.3 × 10^1^ mL^−1^ in late September, a difference of more than one order of magnitude (Figure 2b). In 2012, the number of nitrate reducers raised continuously from 2.1 × 10^1^ mL^−1^ in March to 2.4 × 10^3^ mL^−1^ in October, a difference of two orders of magnitude. In winter, the MPN dropped down again by more than two orders of magnitude from 9.3 × 10^3^ mL^−1^ in December 2013 to 2.3 × 10^1^ mL^−1^ in April 2014 (Figure 2b).

The MPN of denitrifying bacteria increased from minimal 3.6 × 10^−1^ mL^−1^ in April 2011 to 2.0 × 10^2^ mL^−1^ in August (Figure 2b), a difference of almost three orders of magnitude. In 2012, MPN increased again from low values (4.3 and 3.0 mL^−1^ in March and May, respectively) to 2.3 × 10^2^ mL^−1^ in August with a decline afterwards to 2.8 × 10^1^ mL^−1^ in January, a difference of two and one orders of magnitudes, respectively. Similar to the nitrate-reducing bacteria, denitrifyer concentration decreased in winter from 9.3 × 10^3^ mL^−1^ in December 2013 to 2.3 × 10^1^ mL^−1^ in February 2014 (Figure 2b).

The combined results from the three-year study showed a significant seasonality in the MPN of nitrate-reducing and denitrifying bacteria with a maximum in winter and a decrease in spring (Figure 3b). From early summer, the MPN increased to maximum values in August and fall.

The difference between nitrate-reducing and denitrifying bacteria calculated using the MPN approach varied between zero and more than two orders of magnitude without an obvious seasonal course. On average (median), the difference of the log-transformed values was 0.3 log units.

The number of denitrifying bacteria calculated from the sum of *nirS* and *nirK* gene copy numbers was in median 0.79 log units smaller than the 16S cell number. It ranged between 3.7 × 10^4^ (spring 2012) and 2 × 10^6^ mL^−1^ (fall 2012). Similar to the total cell numbers calculated from 16S rDNA, an increasing trend from spring to fall 2012 with a difference of almost two orders of magnitude was detected (Figure 2b), although the values showed large fluctuations. During winter 2013–2014, the number of *nir* gene copies remained on average almost constant (Figure 2b). There were roughly four times more *nirS* gene copies detected than *nirK*.

Combining the results from 2012 to 2014, the *nir* gene copy numbers did not show any significant seasonality (Figure 3b). This was contrary to the significant seasonality in the MPN approach.

The average differences between microscopic and molecular enumeration on the one hand and the MPN values of denitrifyers on the other were roughly 4 log units (Figure 3c). However, the differences showed an opposite seasonal course to the MPN itself with an increase from 3 log units to 5 log units during winter until late spring and a decrease in summer and fall to 3 log units (Figure 3c), which highlights again the seasonal variability of the MPN numbers compared to the stable microscopic and PCR-based cell numbers.

### 3.2. Hypolimnion

In the hypolimnion, the development of the environmental parameters and cell numbers were in parts different to the epilimnion (Figure 1 and Figure 4). The displayed data of temperature, oxygen and nitrate concentrations (Figure 1a–c) reveal high similarities of these parameters between the three studied years. Temperature was slowly increasing during summer and declined after mixing in fall. Oxygen dropped from high concentrations in winter to zero during spring. The nitrate concentrations showed distinct annual courses as well with peak concentrations in winter, a small decline in spring, another peak in early summer and depletion in fall.

As a result of the annual variation in the physical and chemical parameters, the MPN of nitrate-reducing and denitrifying bacteria showed a significant annual course as well. The MPN numbers were highest in winter, declined in spring, and had another maximum in late summer, followed again by a decline in fall until the end of stratification (Figure 4b and Figure 5a).

In detail, the annual course can be described as follows. After a smaller drop in the nitrate concentration in spring, temperature and nitrate concentration increased in June. The MPN numbers of nitrate-reducing bacteria raised correspondingly from 1.1 × 10^2^ to 9.3 × 10^2^ mL^−1^ in 2011 and from 2.1 × 10^1^ to 9.3 × 10^2^ mL^−1^ in 2012 (Figure 4b), a difference of almost two magnitudes. The MPN of denitrifying bacteria increased as well, from less than 1 mL^−1^ in 2011 and 2 mL^−1^ in 2012 to 1.5 × 10^2^ and 2.3 × 10^2^ mL^−1^ in late summer 2011 and 2012, respectively. This was a significant difference of two magnitudes. In late summer and fall, the nitrate concentration dropped and the MPN of nitrate-reducing bacteria also declined to minimum values of 1.5 × 10^2^ mL^−1^ in 2011 and 9.3 × 10^1^ mL^−1^ in 2012 before autumn circulation (Figure 4b). The number of denitrifyers decreased as well to 4.3 mL^−1^ in October 2012. Low values, 4.3 × 10^1^ mL^−1^ for nitrate-reducing bacteria and 2.3 × 10^1^ mL^−1^ for denitrifying bacteria, were also detected in October 2013 (Figure 4b). In winter, the number of nitrate-reducing bacteria reached another maximum of 1.5 × 10^3^ (January 2013) and 9.3 × 10^3^ mL^−1^ (December 2013). This maximum was also detectable for denitrifying bacteria, at least in 2013. In spring 2014, both groups dropped to low values of 4.3 and 2.3 mL^−1^, a difference of more than two magnitudes.

Similar to the epilimnion, the microscopic and 16S rDNA cell numbers in the hypolimnion were almost equal (mean difference of the log-transformed values = 0.1 log units). The *nir* gene copy numbers were on average 0.45 log units smaller than the related 16S rDNA gene copy numbers. Contrary to the epilimnion, the 16S rDNA and *nir* cell numbers showed a similar seasonal trend to the MPN (Figure 4a,b), but with a larger amplitude. The lowest cell numbers were measured in spring 2012 with 1 × 10^5^ mL^−1^ (16S rDNA) and 4 × 10^4^ mL^−1^ (*nir*). During the following stratification, the 16S rDNA and *nir* cell numbers increased up to 1.7 × 10^6^ mL^−1^ and 2.4 × 10^6^ mL^−1^ in August and decreased thereafter until mixing in fall down to 2 × 10^5^ and 4 × 10^4^, respectively. In 2013, the 16S rDNA and *nir* cell numbers reached maximal values of 1 × 10^7^ and 1 × 10^8^ in August, maybe due to higher nitrate concentrations compared to 2012.

Combining all three years, a maximum in summer was always followed by a strong decline to a minimum in fall (Figure 5a). There was another slight 16S rDNA cell-number peak in winter (Figure 5a). Overall, the 16S rDNA cell number showed a variability of two orders of magnitude, the *nir* cell number of 3.5 orders of magnitude. The total microscopic cell counts ranged from 7 × 10^5^ to 5 × 10^6^ mL^−1^ without a detectable seasonal course.

The differences between denitrifying and nitrate-reducing bacteria ranged from zero to 3 log units (average 0.9 log units) without a seasonal dependency. The microscopic and molecular enumerations on one hand and the MPN values on the other differed by roughly 4 log units. This was similar to the epilimnion. Between the *nir* gene copy number and the number of cultivable denitrifying bacteria, a seasonality in the difference can be seen in Figure 5b. In winter and late summer, when the MPN was highest, the difference was lowest (1.2 log units in December, 2.6 in September Figure 5b). At a low MPN in spring 2011 and 2013, but also in October, the difference was highest (4.8 log units).

### 3.3. Sediment

As expected, all bacterial cell numbers in the sediment were much higher than in the pelagial. In 2011, the MPN of nitrate-reducing bacteria were higher in summer (4.8 × 10^5^ mL^−1^) than in November (6.8 × 10^4^ mL^−1^). In 2012, a trend of decline from 2 × 10^6^ mL^−1^ in spring to 4 × 10^4^ mL^−1^ in fall could be observed, a difference of almost two orders of magnitude (Figure 6). The number of cultivable denitrifying bacteria ranged between 2.6 × 10^5^ mL^−1^ in June 2011, 1.5 × 10^4^ mL^−1^ in November 2011 and 2.1 × 10^4^ mL^−1^ in May 2012. A seasonal course could not be deduced. The total microscopic cell numbers were in a range between 8 × 10^7^ and 8 × 10^9^ mL^−1^ with no detectable seasonal course as well (Figure 6). The cell numbers calculated from the 16S rDNA and *nir* gene copy numbers in the sediments were in general similar or up to two orders of magnitude higher than the corresponding microscopic cell counts (Figure 6) and varied between 5.6 × 10^8^ mL^−1^ and 2.5 × 10^11^ mL^−1^. More than 90% of the extracted DNA was not related to intact bacteria. Although the *nir* results suggest higher values between fall and spring and lower values in late summer with a difference of two orders of magnitude, a seasonal dependency is difficult to detect. In general, the *nir* values are similar to the 16S rDNA values (median difference of less than 0.1 log units), although in some cases *nir* genes were higher than 16S rDNA, which eventually is caused by methodical uncertainties.

A statistical analysis revealed some correlations in the data set. The gene copy numbers of 16S rDNA and *nir* were correlated (epilimnion: 0.65, *p* = 0.004, *n* = 17, hypolimnion: 0.57, *p* = 0.006, *n* = 20) as well as the MPN of nitrate-reducing and denitrifying bacteria (epilimnion: 0.69, *p* = 0.02, *n* = 16, hypolimnion: 0.63, *p* = 0.04, *n* = 18). In the hypolimnion from March 2012 to January 2013, the MPNs of nitrate-reducing bacteria and denitrifying bacteria were correlated with *nir* gene copy numbers (0.8, *p* = 0.027 and 0.7, *p* = 0.08, *n* = 7, Figure 5b). Beside this, no further correlation between log-transformed microscopic, PCR-based and MPN cell numbers could be found (*p* > 0.15, *n* = 14 to 18). Significant correlations between log-transformed cell numbers and the environmental factors of temperature, DOC and nitrate could only be found between the MPN of denitrifying bacteria and temperature in the epilimnion (0.65, *p*= 0.01, *n* = 14), when excluding one outlier in December 2013 (Figure 2, otherwise 0.47, *p* = 0.07, *n* = 15), and between the MPN of nitrate-reducing bacteria at the sediment surface and the nitrate concentration of the hypolimnion (0.6. *p* = 0.04, *n* = 10, Figure 6). In addition, from May to October 2012, the MPN of nitrate-reducing bacteria and nitrate concentration in the epilimnion were correlated as well (0.9, *p* = 0.03, *n* = 5, Figure 2).

## 4. Discussion

### 4.1. Comparision with Other Fresh-Water Systems

The large variability of the data hampers a comparison with other studies, if only few samples were analysed throughout the year. The authors of Ref. [52] mentioned an anaerobic MPN of 2 × 10^2^ and 4 × 10^4^ for the epi- and hypolimnion of a lake, and ref. [53] gives a range from of 1 × 10^1^ to 2 × 10^3^. The current MPNs of denitrifying and nitrate-reducing bacteria are within this range. For sediments, the comparison is even more difficult, since PCR-based results depend strongly on the extraction procedure, especially for more complex materials like sediments [54]. In addition, some values are given per mL while others refer to dry weight in grams, often without information for a conversion between both units. Since the DW of the here-studied sediments is between 5 and 6%, one mL would contain about 0.06 g DW or 1 g DW equals to about 17 mL. With this, the MPN range in the sediment is between 2 × 10^4^ and 2 × 10^6^ per mL or about 1 × 10^3^ to 1 × 10^5^ per g DW. The authors of ref. [52] reported a value of 1 × 10^9^ per mL, while, for example, the author of ref. [10] gives a range from 3 × 10^4^ to 6 × 10^5^ per g DW, which is similar to our results, and the authors of ref. [55] mentioned a range from 2.3 × 10^6^ in October to 1.2 × 10^7^ in August, which is one order of magnitude larger, yet with similar seasons for the minimum and maximum values like the current annual course for the hypolimnion.

For the *nir* gene copy number, the authors of [39] reported peak concentrations of about 7 × 10^5^ mL^−1^ in the hypolimnion in summer and minimum concentrations in fall and winter below 1x10^5^, which is less than our findings but with a similar seasonality. For the sediment, our measured range of *nir* gene copy numbers is higher than other reports, but corresponds with the microscopic values. The authors of ref. [56] reported a *nir* gene copy number of 2 × 10^6^ probably per gram fresh weight and [57] a concentration of 1 × 10^6^ g^−1^ DW, while refs. [39,58] measured a concentration of about 2 × 10^8^ per gram DW without seasonality, and the authors of ref. [59] measured about 2 × 10^7^ per gram wet weight.

### 4.2. Seasonal Course of Bacteria and Dependency on Environmental Parameters

The seasonal courses of denitrifying and nitrate-reducing bacteria in different compartments of a dimictic lake have been measured using different methods. As shown, the seasonality was best displayed by the cultivation-based MPN technique. Although this method has a high inherent statistical error, significant differences of more than two orders of magnitude could be detected within the annual courses in the epi- and hypolimnion. The seasonal courses of the cell numbers differed between epilimnion, hypolimnion and sediment, due to different driving environmental parameters.

In the epilimnion, the MPN of nitrate-reducing and denitrifying bacteria increased from minimal values in spring at low temperatures until fall to up to two orders of magnitude at parallel increasing temperatures and increasing nitrate concentrations. Although both parameters are correlated with the MPN at this time, the growth during summer seems to be driven by temperature and oxygen rather than by the availability of nitrate, since many denitrifying bacteria prefer oxygen, which was present throughout the year. In winter at low temperatures, the MPN of nitrate-reducing bacteria dropped back to minimal values.

DOC as another major factor for heterotrophic bacteria was fairly constant at limiting values between 6 and 8 mg∙L^−1^, much lower than the K_m_-value (for example about 70 mg L^−1^ [60]. Thus, it could not be proven to be responsible for the seasonal cell number variation, but will certainly limit the maximum cell number at otherwise favourable conditions. However, the increase of the MPN in summer and fall might at least partially be influenced by a higher carbon supply from growing and degrading phytoplankton. This can be concluded from the reported variation in the chlorophyll concentration [41]. Here, the combination of increased supply, immediate consumption and increasing bacterial concentrations would keep the DOC values at constant low concentrations.

The described seasonal course in the hypolimnion was later confirmed by a more detailed sampling campaign in 2015 [61]. In contrast to the epilimnion, in the hypolimnion, nitrate was the main electron acceptor for denitrifying bacteria, since oxygen was depleted very fast in spring in part by nitrification [61]. At increasing temperature and nitrate concentrations, the MPN cell numbers of nitrate-reducing bacteria in the hypolimnion also increased by almost two orders of magnitudes from spring until summer and fell back to low values parallel to nitrate depletion in late summer. However, no direct correlation could be found between denitrifying bacteria and environmental parameters. This is most likely caused by an offset between environmental conditions and the slow responses of the cell numbers in the colder hypolimnion. This can be seen, e.g., in the delay of the MPN compared to the course of nitrate in the combined annual course (Figure 5) and in proceeding studies [61]. Although the nitrate concentration is already declining, it is still sufficient to support continuous bacterial growth until depletion.

For the upper layer of the sediment, similar conclusions can be drawn. The number of nitrate-reducing bacteria was correlated to the nitrate concentration and declined by almost two orders of magnitude in summer at depleting nitrate concentrations.

### 4.3. Explanations on the Seasonal Courses

The seasonal course of the *nir* cell number in the epilimnion was less pronounced than the MPN. In summer 2012, it increased by 1.8 orders of magnitude compared to two and three in the MPN. At impairing environmental conditions and low temperatures, the MPN significantly decreased while the *nir* cell number remained constant (best to see in winter 2014). As a result, the differences between *nir* gene copy numbers and the MPN of and nitrate-reducing and denitrifying bacteria, respectively, varied between 1 and 1.9 log units at the end of favourable conditions and up to 3 and even 5 orders of magnitude at impairing conditions. This observation can be explained at least in two different ways.

A first explanation could be that only a fraction of all nitrate-reducing and denitrifying species are cultivable under artificial laboratory conditions, thus causing the difference between the MPN and *nir* gene copy numbers in front of a large non-cultivable background of eventually similar numbers like the *nir* cell number. If, moreover, it is assumed that bacteria multiplied under favourable conditions and died at non-favourable conditions, the multiplication of cultivable cells would only slightly affect the *nir* cell number on a logarithmic scale, due to the large difference between MPN and *nir* cell numbers at the starting point. However, in Lake Scharmützelsee, the ratios between the *nir*-gene abundance and the number of cultivable bacteria were changing by more than two orders of magnitude during the seasonal course. To interpret these findings in the context of the explanation above, a faster growth of cultivable species compared to non-cultivable cells would be necessary. Such a biased detection of cultivable r-strategists in front of an unculturable background of k-strategists seems possible. On the other hand, mainly active cultivable cells need to die at non-favourable conditions in winter to cause the increasing difference between *nir* cell numbers and MPN.

The multiplication and degradation of cells in front of a larger pool of cell-free intact DNA or the DNA of dead cells seems less likely. The DNA has been degraded in early spring parallel to constant or even increasing microscopic cell counts as can be concluded from the decreasing 16S rDNA gene copy numbers.

A second explanation seems more likely. We assume that a large portion of denitrifying bacteria change from a viable into a dormant or VBNC state [62] during non-favourable conditions (e.g., at cold temperatures as seen in spring 2014). As a result, the MPN counts varied strongly between adequate and reverse conditions within the annual course, whereas the cell numbers detected by molecular or microscopic methods showed less seasonal variability.

According to our assumption, dormant denitrifying bacteria become active again at favourable conditions as discussed in several other studies before [62]. This explanation is supported by laboratory tests reported by the authors of [24] with pure cultures of denitrifying *Pseusdomonas* at growing and starving conditions. Alternatively, several investigators also discussed the possibility that a relative small number of cells remained vital under unfavourable conditions or become active independent of environmental factors and start to multiply at adequate environmental conditions [63]. This would also result in a relatively smaller increase in the *nir* gene copy number due to the large background.

Another explanation, the formation of micro-colonies at non-favourable conditions, as assumed by the authors of [64] for soil-nitrifying bacteria, can be excluded. The microscopic analysis did not reveal those clusters at least in pelagic samples.

Combining the results from the cited laboratory tests and this study, it can be argued, that the often-reported large difference between cultivation-based and other cell-counting methods (e.g., [29]) are caused at least in part by a proportion of cells in the VBNC or dormant state and not exclusively by the occurrence of non-cultivable species. At the end of growing periods, cell counts obtained either from MPN or gene copy numbers matched quite well.

In contrast to the epilimnion, the seasonal increase and decrease in the MPN, *nir* and 16S rDNA gene copy numbers in the hypolimnion were comparable. This could be explained by a higher feeding pressure by bacterivourus placton like ciliates and flagellates in the hypolimnion compared to the epilimnion, as reported by the authors of [65]. Feeding would deduce a non-active and therefore non-multiplying bacteria population including their DNA resulting in only low amounts of dormant or VBNC cells and a parallel course of MPNs and *nir* cell numbers.

For the sediment, it is difficult to draw conclusions. There was a decline in the *nir* gene copy number and the MPN from December 2011 to summer 2012 at decreasing nitrate concentrations while the 16S barely changed. The death and eventually lysis of denitrifying cells and a switch between active and dormant cells could not be distinguished, since the total viable cell number was several orders of magnitudes higher than the number of denitrifying ones. However, similar to the pelagial, the MPN values reflected the changing environmental conditions best.

In the pelagial, there was a good match between microscopic and molecular total cell numbers with generally less than one magnitude difference. This can be explained by a low amount of cell-free intact genetic material. Therefore, both methods seem to be appropriate to estimate the total cell number in pelagic environments and other systems with a fast degradation of free DNA. The *nir* gene copy number reflects the total amount of denitrifying cells independent of their cultivable status.

In contrast to the pelagial, microscopic cell numbers and gene copy numbers in the sediment differed by at least one order of magnitude. One reason might be a possible underestimation of cells in sediment particles, although this effect proved to be very small [51]. Another more likely explanation is a high abundance of intact DNA in dead cells or extracellular DNA bound to particles. Previous studies reported that the amount of extracellular DNA adsorbed to sediment particles can be 10 to 70 times higher than the cellular DNA concentrations [66]. In those environments with a slow degradation of free DNA, the estimation of cell abundance by measuring the genomic gene copy number using calibration curves made by log-phase cultures [67]) will lead to an overestimation by eventually several orders of magnitude. These limits of molecular techniques for measuring active metabolizing cells have been already described by other investigators before. The authors of ref. [68] demonstrated that the success of disinfection could be monitored by the depletion of cultivable cells while the number of gene copies only slightly decreased. The authors of ref. [11] described a correlation between gene copies and potential enzymatic activity for only some reactions in glacier soil and argued with inactive populations due to substrate shortage on one side and with uncertainties in the potential activity measurement due to shifting population on the other. The discrepancies in the correlation of *nir* gene copies and denitrification activities reported by the authors of [5] might in part also be explained by an inactive portion of genetic material. Here, other molecular techniques like the detection of m-RNA [69] seems a better choice, since its half-time is short [70]. However, the m-RNA amount per cell can change depending on starvation and resuscitation [71] and cannot directly be related to metabolic activity [72]. As an alternative, several investigators tried to separate extracellular DNA during the extraction procedure [66].

In general, as shown in this paper, a combination of both cultivation and molecular techniques or microscopic cell counts seems necessary as recommended earlier [22,30] and will be helpful for the interpretation of the results especially in sediments.

## 5. Conclusions

This study demonstrates that the population of active denitrifying bacteria in a dimictic lake can change by several orders of magnitude within a seasonal course. There are differences between the epi- and hypolimnion due to different driving factors, which are mainly organic carbon and temperature in the epilimnion and nitrate in the hypolimnion and sediment. To estimate the seasonal variability of the active metabolizing cell number especially in highly dynamic systems with changing environmental conditions like nutrient or oxygen concentrations, cultivation-based techniques might give a better understanding than classic molecular approaches. These results can be of particular importance, when using the cell number for the calculation of in situ metabolic activities and turnover rates [61]. On the other hand, to estimate the growth dynamic of a bacterial population, the quantification of the gene copy number might be an important tool as well, if a major portion of the cells simply switch between dormant and active conditions, because the MPN number will increase without the multiplication of cells. However, this will only be suitable for environments with a small background of cell-free DNA or the DNA of dead cells. The study focused on a deep dimictic lake. A future comparison with other lake types or fresh-water systems would be interesting to improve the understanding of the complex bacterial population dynamic in aquatic systems, since this is directly linked to the nitrogen cycle and the role of those systems as a nitrogen sink.

## Figures and Tables

**Figure 1 microorganisms-12-00511-f001:**
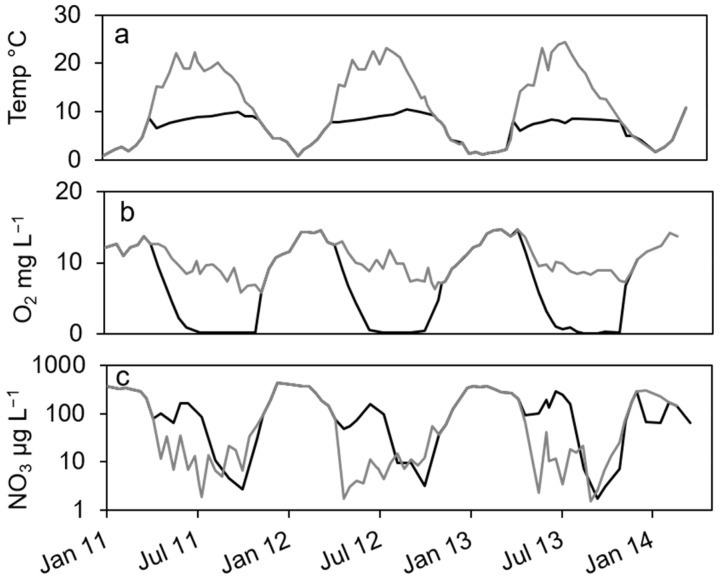
Courses of temperature (**a**), oxygen (**b**) and nitrate concentration (**c**) in the epilimnion (grey line) and hypolimnion (black line) of lake Scharmützelsee from 2012 to 2013.

**Figure 2 microorganisms-12-00511-f002:**
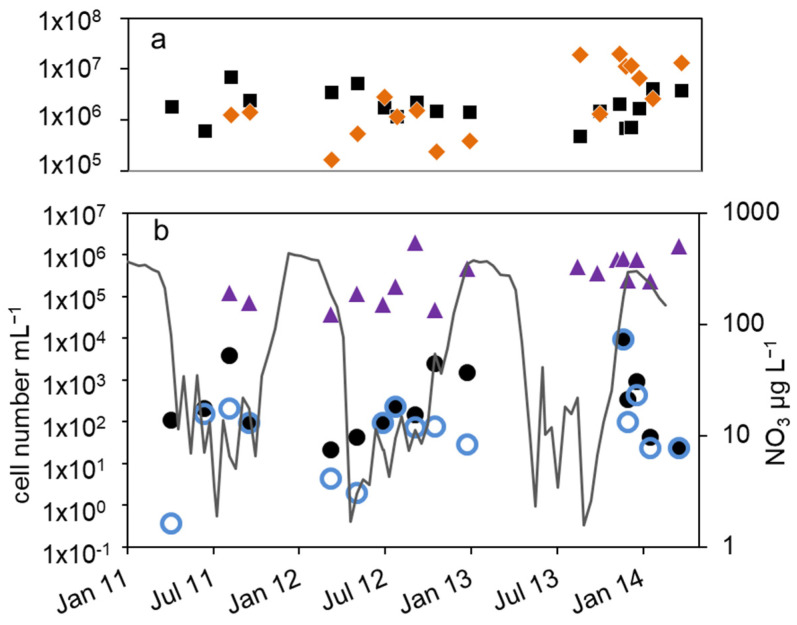
Courses of nitrate concentration and cell numbers in the epilimnion of lake Scharmützelsee from 2012 to 2014: (**a**) microscopic cell counting (
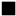
), 16S rDNA gene copy number (
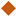
), (**b**) MPN of nitrate-reducing bacteria (
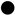
) and denitrifying bacteria (
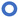
), sum of *nirS* and *nirK* (
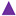
), nitrite concentration (grey line).

**Figure 3 microorganisms-12-00511-f003:**
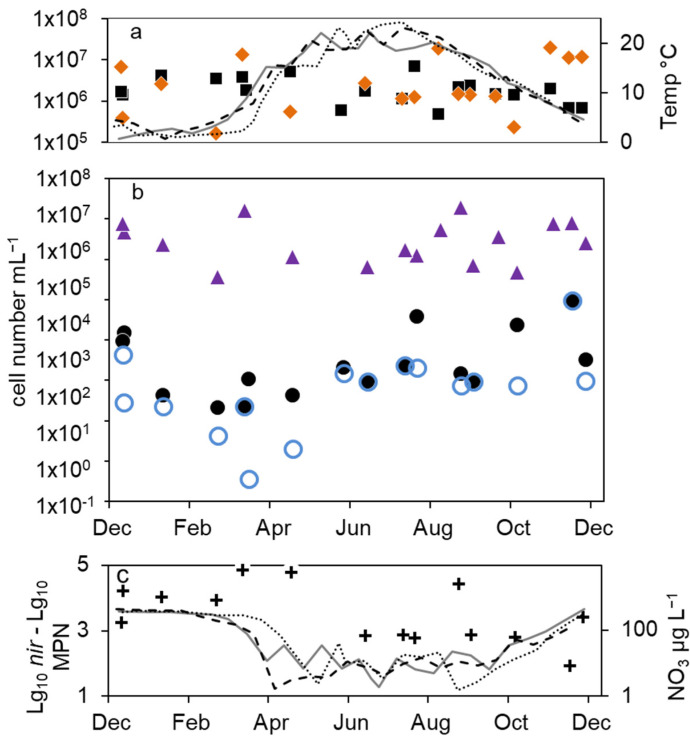
Combined annual courses of cell numbers and nitrate concentrations in the epilimnion of lake Scharmützelsee: (**a**) microscopic cell counts (
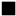
), 16S rDNA cell-number estimation (
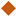
), temperature 2011 (solid grey line), 2012 (dashed black line), 2013 (dotted black line) (**b**) numbers of denitrifying bacteria estimated by the sum of *nirS* and *nirK* gene copies (
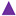
), MPN of nitrate-reducing bacteria (
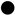
) and denitrifying bacteria (
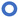
), (**c**) difference between log10 transformed cell numbers (nir)and MPN of denitrifying bacteria (+), nitrate concentration 2011 (solid grey line), 2012 (dashed black line), 2013 (dotted black line).

**Figure 4 microorganisms-12-00511-f004:**
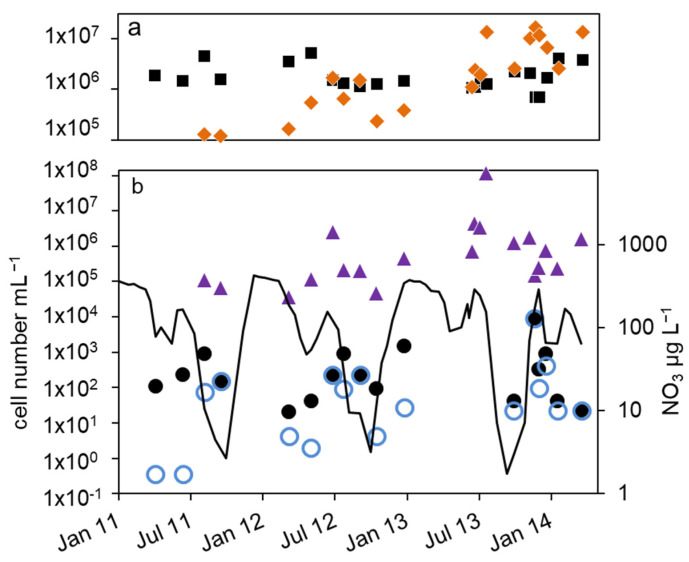
Courses of nitrate concentration and cell numbers in the hypolimnion of lake Scharmützelsee from 2012 to 2014: (**a**) microscopic cell counting (
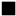
), 16S rDNA cell-number estimation (
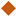
), (**b**) number of denitrifying bacteria estimated by the sum of *nirS* and *nirK* gene copies (
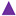
), MPN of nitrite-reducing bacteria ((
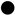
)) and denitrifying bacteria (
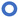
), nitrate concentration (black line).

**Figure 5 microorganisms-12-00511-f005:**
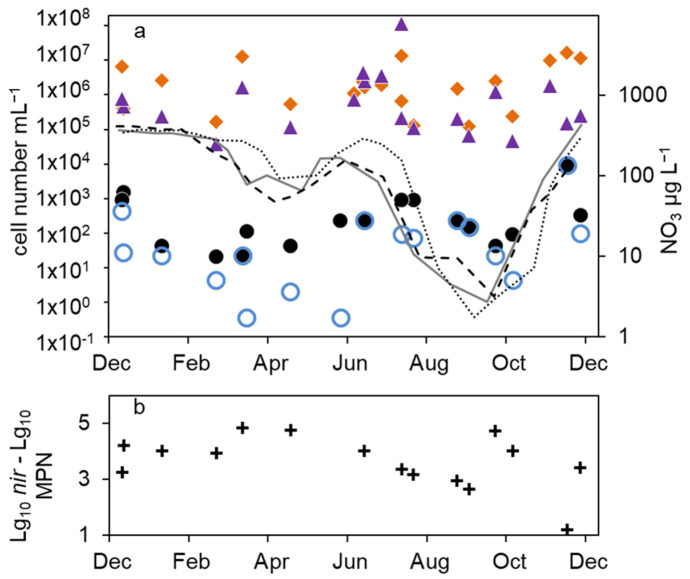
Combined annual courses of cell numbers and nitrate concentrations in the hypolimnion of lake Scharmützelsee: (**a**) 16S rDNA cell-number estimation (
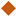
), number of denitrifying bacteria estimated by the sum of *nirS* and *nirK* gene copies (
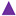
), MPN of nitrate-reducing bacteria (
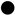
) and denitrifying bacteria (
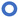
), nitrate concentration 2011 (solid grey line), 2012 (dashed black line), 2013 (dotted black line), (**b**) difference between log10-transformed cell numbers (nir) and MPN of denitrifying bacteria (+).

**Figure 6 microorganisms-12-00511-f006:**
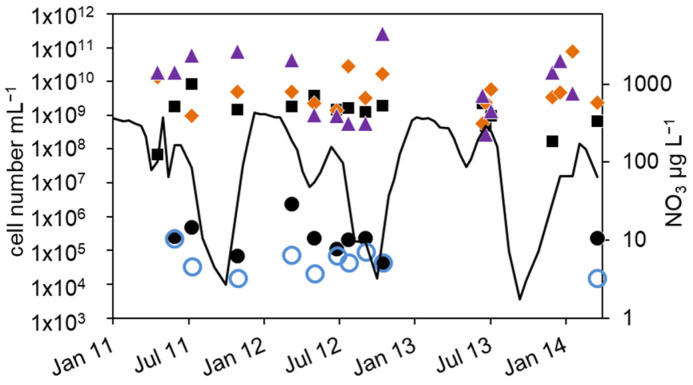
Courses of nitrate concentration in the hypolimnion and cell numbers in the upper sediment layer of lake Scharmützelsee from 2012 to 2014: microscopic cell counting (
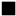
), 16S rDNA cell-number estimation (
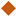
), number of denitrifying bacteria estimated by the sum of *nirS* and *nirK* gene copies (
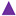
), MPN of nitrite-reducing bacteria (
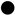
) and denitrifying bacteria (
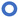
), nitrate concentration (black line).

## Data Availability

Raw data on cell numbers and physico-chemical parameters are available at https://doi.org/10.13140/RG.2.2.28332.85122/1 (accessed on 27 February 2024).

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
