# Peer review of "Seasonal Variability of Cultivable Nitrate-Reducing and Denitrifying Bacteria and Functional Gene Copy Number in Fresh Water Lake"

_microorganisms, 2024, doi:10.3390/microorganisms12030511_

Round 1
Reviewer 1 Report
Comments and Suggestions for Authors
In this study, the authors analyzed the seasonal dynamics of the total microbial population and those involved in denitrification and nitrate reduction by a combination of cultivation-dependent and PCR-based techniques. The goal was to compare the results of those two approaches for the interpretation of seasonal dynamics.
In general, the approach to compare the total denitrifier community to that fraction which is presumably active is valuable. However, I have several concerns regarding the methodological approach used in this study, the structure of the manuscript, and the interpretation of the results.
(i) The PCR approach for the quantification of functional genes appears a bit uncommon given that quantitative PCR is widely used these days. Comparing the active community to a total community which was assessed with potentially inaccurate methods is weakening the whole scientific interpretation. The authors should at least check some of their samples on a standard SYBR Green based qPCR protocol to test how well these different quantification methods agree.
(ii) Throughout the manuscript, discussion of results of the total microbial community (“cell numbers”) and the denitrifier and nitrate reducer community is often not well separated so that it is not clear to the reader if the authors refer to the total microbial community or their functional communities of interest. Here, rephrasing and restructuring in multiple places is needed to improve this distinction.
(iii) In general, the manuscript is not easy to follow as throughout the different sections, the authors combine the description of results and observations with the evaluation of the differences in results obtained with different methods. This makes the manuscript hard to read. I would suggest that the authors separate these two layers more clearly. This way, the reader can first get a good understanding of the system and in the second step learn about the difference and evaluation of the different enumeration methods.
(iv) The authors state that cultivation-based techniques might give a better understanding than molecular approaches when it comes to the analysis of seasonal dynamics of metabolically active cells. However, the authors should also keep in mind that cultivation bias may also lead to an underestimation of the metabolically active cells because they might need different growth conditions than what is offered in the laboratory incubations.
(v) In general, the discussion is very much focused on the lake system studied here and needs the integration of more literature to allow some generalizations and comparisons of findings with other studies.
Specific comments:
l. 60 and in other places: dead DNA? Rather rephrase as DNA from dead cells
l. 83-92: I would suggest adding a clear objective, research question or hypothesis here. Is the work mostly method-oriented or do the authors also aim to understand nitrogen cycling processes in their study system? This is not totally clear from this section.
l. 108-140: This is rather a description of results than a method description and should be placed in the results section.
l. 243 and in other places: “16S cell number” is not a very accurate term. Please rephrase as “cell numbers estimated based on 16S rRNA gene abundances”
l. 434: This Km value may not apply to all environmental heterotrophic bacteria.
l. 505-510. This appears very speculative. Why would grazers preferentially feed on non active or VBNC cells?
l. 572: Should it not be “active denitrifying bacteria” here?
Figure 2 and 3: There seems to be an error regarding the color code for 16S rDNA cell number estimation, please check.
Author Response
Dear Reviewer,
Thank you very much for the detailed critics on our manuscript.
- The PCR approach for the quantification of functional genes appears a bit uncommon given that quantitative PCR is widely used these days. Comparing the active community to a total community which was assessed with potentially inaccurate methods is weakening the whole scientific interpretation. The authors should at least check some of their samples on a standard SYBR Green based qPCR protocol to test how well these different quantification methods agree.
We agree, that the PCR quantification method we used is meanwhile outdated. We acknowledged this a little in line 55 and 56. To cover the uncertainties of all used methods we specifically included an entire paragraph on this, which covers the significant differences of all methods. I addition the data are supported by microscopical counts and internal controls. The manuscript is quite large already, a detailed comparison and discussion of different molecular approaches including the difficulties and variability of extraction procedures, which is a crucial and a very vulnerable part of PCR techniques (see Riedel at al 2017 DOI: 10.1016/j.mimet.2017.10.007, will expand the manuscript most likely too much. Jet, we added some statements to that issue and the mentioned the reference.
For the seasonal course we used the same method for the entire campaign to ensure the consistency for each individual method. We agree, that the comparison between different parameters whith individual variabilities leads to increasing uncertainties. However, that is a problem of all comparisons and correlations between slightly imprecise parameters. Jet the differences between the parameters are reasonably consistent over three years.
(ii) Throughout the manuscript, discussion of results of the total microbial community (“cell numbers”) and the denitrifier and nitrate reducer community is often not well separated so that it is not clear to the reader if the authors refer to the total microbial community or their functional communities of interest. Here, rephrasing and restructuring in multiple places is needed to improve this distinction.
Changes made: Line 228, 263
In the paragraph 324 to 343 we skipped the specification “MPN” or similar explanations, since the entire paragraph discusses this group, which we believe is quite obvious.
Changes made: line 344 and following, for example line 377, 391, 392, 402
(iii) In general, the manuscript is not easy to follow as throughout the different sections, the authors combine the description of results and observations with the evaluation of the differences in results obtained with different methods. This makes the manuscript hard to read. I would suggest that the authors separate these two layers more clearly. This way, the reader can first get a good understanding of the system and in the second step learn about the difference and evaluation of the different enumeration methods.
We appreciate the honest comment. We are aware of the difficulties of the manuscript since it is a very complex content. Of course, there are different ways to structure the content. However, we think, that another structure will have drawbacks as well. Separating the pure description from eventually important comparisons between the different parameters too much as suggested will probably cause redundancies or the author has to switch a lot between different paragraphs. Jet we made small changes.
(iv) The authors state that cultivation-based techniques might give a better understanding than molecular approaches when it comes to the analysis of seasonal dynamics of metabolically active cells. However, the authors should also keep in mind that cultivation bias may also lead to an underestimation of the metabolically active cells because they might need different growth conditions than what is offered in the laboratory incubations.
Thank you very much for this comment. Actually, we thought a lot about that issue und tried to cover it at least a little in line 471 and following. We extended the discussion on this issue slightly. However, a deep discussion on that issue would fill an entire publication so we hope, our discussion is sufficient for this already complex manuscript.
(v) In general, the discussion is very much focused on the lake system studied here and needs the integration of more literature to allow some generalizations and comparisons of findings with other studies.
Thank you. Interestingly, many papers of that topic do not compare with other studies. We added an entire paragraph on this issue.
Intentionally or unintentionally the reviewer made us this way noticing a mistake in the conversion between Liter an mL for the PCR based quantification in the sediments. We corrected the valued and adapted the result and discussion section accordingly. Thank you.
Specific comments:
- 60 and in other places: dead DNA? Rather rephrase as DNA from dead cells
done
- 83-92: I would suggest adding a clear objective, research question or hypothesis here. Is the work mostly method-oriented or do the authors also aim to understand nitrogen cycling processes in their study system? This is not totally clear from this section.
We have modified the last paragraph of the introduction.
- 108-140: This is rather a description of results than a method description and should be placed in the results section.
Yes, we thought about that issue. However, since the focus of the manuscript is on the microbiology we would like to keep the description of the abiotic parameters as a general system and sampling site description in the method section.
- 243 and in other places: “16S cell number” is not a very accurate term. Please rephrase as “cell numbers estimated based on 16S rRNA gene abundances”
we made changes
- 434: This Km value may not apply to all environmental heterotrophic bacteria.
That is certainly true but a detailed discussion could also easily fill at least an entire paragraph. We are certain, that the statement is sufficient to show, that DOC is not the main driving parameter for the seasonality. We changed slightly.
- 505-510. This appears very speculative. Why would grazers preferentially feed on non active or VBNC cells?
I guess our statement was misinterpreted. We argue that a dormant population at grazing pressure is declining including the gene copy number as seen in the data. There is no selective grazing. We clarified the sentence.
- 572: Should it not be “active denitrifying bacteria” here?
OK, thank you. that would be more precise.
Figure 2 and 3: There seems to be an error regarding the color code for 16S rDNA cell number estimation, please check.
Changed, Thank you.
Further changes:
We noticed a little left over from the Style sheet in the Manuscript. It is deleted.
304 to 304: some clarification added
Line 344, slight changes in the structure
Line 404: sentence deleted
Line 469: explanation extended for a better understanding
Line 511: changes made for clarification
Line 297: sentence deleted after reconsidering the variability in the data
Line 302: rephrased for clarification
Line 552: slight changes in the structur
Line 595: we deleted a redundant paragraph
Reviewer 2 Report
Comments and Suggestions for Authors
On manuscript on Böllmann & Martienssen on the role on cultivable methods versus molecular methods on the spur on denitrifying bacteria on Lake Scharmützelsee. Although manuscript carries relevant findings on the seasonal variation on this particular ecosystem on lacks a discussion on application on their findings on other ecosystems. On began authors should provide an overview on the characteristics on choosing Lake Scharmützelsee on their spurs? Another remark falls on the lack on a limiting parameter on explaining the seasonal variation? I mean what causes the shifts on denitrifying bacteria on this lake or on any other lakes? Authors should refer on other studies on this matter. Also, on chosen methods on main novelty on comparison on other studies? On molecular methods versus culturable methods the measurement on viable cells versus nonviable cells on relevant on spurring environmental bacteria? Authors should provide a short overview on the advantages on each method and compare on other findings.
Finally, on environmental implications on the findings on Lake Scharmützelsee? Any point or nonpoint sources on pollution could be contributing on data?
Author Response
Dear reviewer, Dear Editor,
Thank you very much for your comments.
On manuscript on Böllmann & Martienssen on the role on cultivable methods versus molecular methods on the spur on denitrifying bacteria on Lake Scharmützelsee. Although manuscript carries relevant findings on the seasonal variation on this particular ecosystem on lacks a discussion on application on their findings on other ecosystems.
Responds: Few additional statements are made. We stated in the conclusion and discussion, that the approach and the results can be applied for the calculation and temporal and local specification of denitrification processes, which will apply not only to this specific lake but to aquatic ecosystems in general.
On began authors should provide an overview on the characteristics on choosing Lake Scharmützelsee on their spurs?
Responds: We added a small statement. We already mentioned that main parameters show typical annual courses.
Another remark falls on the lack on a limiting parameter on explaining the seasonal variation? I mean what causes the shifts on denitrifying bacteria on this lake or on any other lakes? Authors should refer on other studies on this matter.
Responds: Actually, we wrote quite a lot about the limiting conditions or main responsible parameters for all three compartments (chapter 4.1). We added some additional statements.
We have cited already plenty of other studies in the introduction, which deal with different aspects of our study, but since the first reviewer made similar suggestions, we added an entire paragraphand cite references with similar argumentations.
Also, on chosen methods on main novelty on comparison on other studies? On molecular methods versus culturable methods the measurement on viable cells versus nonviable cells on relevant on spurring environmental bacteria? Authors should provide a short overview on the advantages on each method and compare on other findings.
We added some additional statements (Line 491 for example) but have covered advantages and problems of the methods several times throughout the text. A deep discussion on this would fill an entire paper. We hope with the few additions it is sufficient.
Finally, on environmental implications on the findings on Lake Scharmützelsee? Any point or nonpoint sources on pollution could be contributing on data?
The history and detailed characterization of the lake is covered in a cited reference and is in our opinion not much relevant for the seasonality, since there are no artificial peak loads documented.